# Pediatric Vulvar Lichen Sclerosus—A Review of the Literature

**DOI:** 10.3390/ijerph18137153

**Published:** 2021-07-04

**Authors:** Dominika Orszulak, Agnieszka Dulska, Kacper Niziński, Kaja Skowronek, Jakub Bodziony, Rafał Stojko, Agnieszka Drosdzol-Cop

**Affiliations:** Chair and Department of Gynecology, Obstetrics and Oncological Gynecology, Medical University of Silesia in Katowice, Markiefki 87, 40-211 Katowice, Poland; dulska.agnieszkaz@gmail.com (A.D.); kacpern@op.pl (K.N.); skowronek.kaja@gmail.com (K.S.); jt.bodziony@gmail.com (J.B.); rafal@czkstojko.pl (R.S.); cor111@poczta.onet.pl (A.D.-C.)

**Keywords:** vulvar lichen sclerosus, pediatric, adolescent

## Abstract

Vulvar lichen sclerosus (VLS) is a chronic inflammatory condition affecting the anogenital region, which may present in a prepubertal or adolescent patient. The most popular theories are its autoimmune and genetic conditioning, although theories concerning hormonal and infectious etiology have also been raised. The most common presenting symptoms of VLS is vulva pruritus, discomfort, dysuria and constipation. In physical examination, a classic “Figure 8” pattern is described, involving the labia minora, clitoral hood, and perianal region. The lesions initially are white, flat-topped papules, thin plaques, or commonly atrophic patches. Purpura is a hallmark feature of VLS. The treatment includes topical anti-inflammatory agents and long-term follow-up, as there is a high risk of recurrence and an increased risk of vulvar cancer in adult women with a history of lichen sclerosus. This article reviews vulvar lichen sclerosus in children and provides evidence-based medicine principles for treatment in the pediatric population. A systematic search of the literature shows recurrence of VLS in children. Maintenance regimens deserve further consideration.

## 1. Introduction

Vulvar lichen sclerosus (VLS) is a chronic inflammatory disease of unclear etiology. The most popular theories are its autoimmune and genetic conditioning, although theories concerning hormonal and infectious etiology have also been raised [1]. VLS manifests in lesions in vulvar mucosa, which often spreads to the skin of the anus [1]. The symptoms of this condition may include whitening of the perineal area, but also itching, burning, discomfort, vaginal bleeding, and dysuria, which in sexually active girls may be mistaken for symptoms of urogenital infection [2]. In some cases, due to anorectal lesions, additional symptoms may occur, such as constipation or painful defecation, without any gastrointestinal problems in the patient’s medical history [3]. The prevalence of VLS in underaged girls is understated, due to the misreading of the symptoms by GPs (general practitioners), and the delayed access to specialists in the field of pediatric gynecology or dermatology [4]. Improving the knowledge about this disease will allow us to start the diagnostic process much earlier, and also to introduce the appropriate treatment more quickly [5,6]. In the literature, there is good evidence for the use of high-potency corticosteroids as the initial treatment, but the maintenance treatment and most effective long-term management strategies are not established. Further, well-conducted randomized controlled trials, with long-term follow-up in the pediatric population are required in order to establish VLS treatment.

## 2. Epidemiology

VLS can occur at any age or in any sex, although the highest values can be observed in women aged 40–60 years old, and in pre-pubertal girls. There is a clear peak of incidence in girls aged four to six years old, which represents 7–15% of all vulvar lichen sclerosus cases [7].

It is estimated that VLS can be observed in 1:900 of premenarchal girls [4,8]. The first symptoms are usually very non-specific and misdiagnosed by non-gynecologist and non-dermatologist doctors. Some of the symptoms can spontaneously recede after the menarche, and the course of the disease can be latent. This is why the epidemiology of VLS is underestimated [8].

## 3. Etiopathogenesis

The etiopathogenesis of VLS remains unknown and is probably multifactorial. There are multiple theories regarding the potential etiopathogenesis of vulvar lichen sclerosus.

### 3.1. Immunological Theory

The available data put emphasis on the role of immunological factors. Even the first case of VLS, reported by Hallopeau, indicated the potential link with diseases such as scleroderma [9]. Recent studies show that 15–34% of cases in adult women and 14% in girls coexist with allergies or autoimmune diseases, such as the following: vitiligo, thyroiditis, type 1 diabetes mellitus, alopecia areata, or celiac disease [1].

The typical lymphocytic cell infiltration may suggest the involvement of interleukin-1 (IL-1) and the antagonist of the IL-1 receptor. It is also considered that some types of human lymphocyte antigens are connected with both a higher or lower risk of vulvar lichen sclerosus development [10].

A variety of studies also report an association with autoimmune diseases such as psoriasis, lichen planus, or morphea. Simpkin et al. found that 48% of patients with VLS present active tissue autoantibodies, whereas thyroid disorder was found in 19% of patients. What is more, only 5% of the patients in this study had a family history of VLS. This is why patients should be screened for autoimmune diseases, although the specific antibody that can become a marker of VLS has not yet been found [11,12].

### 3.2. Genetic Theory

Other researchers suggest the potential role of genetic susceptibility. According to Powell et al. (2001), the family history of autoimmune disease appears to be very visible in the early onset group [4]. In the Sherman et al. work, 12% of female VLS patients had a first-degree female relative who had also been diagnosed with VLS. Numerous case reports of twin, sibling and mother–daughter relationships contribute to a genetic factor in the etiology of lichen sclerosus et atrophicus [13]. This may have implications for future treatment, and for the prediction of the disease [14]. The literature also includes other genetic factors, i.e., Turner or Down syndrome [15,16].

### 3.3. Hormonal Theory

One of the most popular theories concerning the etiology of vulvar lichen sclerosus was also the hormonal background of the disease, especially the theory that suggests hypoestrogenism as a risk factor. The two peaks of incidence, one in pre-pubertal and the other in postmenopausal women, indicate the possible connection with low levels of estrogen during those periods of their life. However, there are not enough studies to prove this theory. In the past, there were also attempts to treat VLS with testosterone, as it was assumed that there was a local testosterone deficiency in the affected tissues [17]. To date, there is also not enough evidence proving this theory. The hormonal role seems to be disputable, and its significance has been falling in recent years. The increased risk of VLS in Turner syndrome, despite the use of HRT (hormone replacement therapy) and contraceptive pills, points to the disputable role of hormones [15].

### 3.4. Trauma Theory

At this point, it is also worth mentioning other factors that are considered to be predisposing to VLS. Some authors take into account local components. One of the well-established signs of vulvar lichen sclerosus is the Koebner response. If the skin is traumatized by a physical factor, lesions following the line of trauma appear [15]. Therefore, repeatable irritation is believed to be a possible factor contributing to the disease. Another possible causal factor may be radiation [18,19,20].

### 3.5. Infectious Theory

Moreover, it should be mentioned that the infectious etiology of vulvar lichen sclerosus was considered multiple times in the past. Although there was much interest in Borrelia burgdorferi and human papilloma virus (HPV), those theories were later disproved [21].

Recently, it was noted that the bacterial environment of the skin and intestines may contribute to the development of the condition. Chattopadhyay et al., in a pilot small-sample study (*n* = 13), showed promising results in this matter. They suggest that there may be a relation between intestinal and cutaneous dysbiosis, and the disease in children, although this needs further investigation [22].

### 3.6. Drug-Induced Theory

There is not much evidence about drug-induced VLS, although some authors report that carbamazepine and imatinib may contribute to the disease. Yet, interestingly, imatinib is an inhibitor of the tyrosinokinase in the cells affected by the condition, and its use is being investigated as a potential treatment method [23,24]. On the other hand, some hypertension drugs, such as ACE inhibitors (angiotensin-converting enzyme inhibitors) or beta-blockers show an inverse relationship with VLS [25].

## 4. Symptoms

The vulvar lichen sclerosus course is very heterogeneous; therefore, it causes diagnostic difficulties. The first symptoms are usually very non-specific and misdiagnosed by non-gynecologist and non-dermatologist doctors, thus, the time between the first examination and the definitive diagnosis may be prolonged over the course of many years [26]. Lagerstedt et al. (2013) claim that only 16% of those with VLS are diagnosed in the early stage of the disease [27]. The average age of symptom onset in girls with vulvar lichen sclerosus is 7.1 years. The average delay from symptom onset to diagnosis is 1.3 years [28].

The most common symptoms that are reported by the patients are itching, edema and a burning sensation of the vulva, accompanied by perineal pain, vaginal bleeding, dysuria, and constipation [4,29]. Further, 86% of the patients report pruritus. This particular symptom often exacerbates in the late evening hours, leading to children’s exhaustion during the day. The excessive genital rubbing can lead to tearing the delicate skin, which can result in bleeding. Some of the patients present an asymptomatic course of the disease. In 30% of girls with VLS, the definitive diagnosis is delayed by ongoing vulvar infections [1,2].

In physical examination, we can detect clearly demarcated white skin lesions, which have a characteristic shape that can be compared to the “Figure 8” or the hourglass (lesions involving the labia minora, clitoral hood, and perianal region). The skin on the labia majora, clitoris, and in the anal area is atrophic, smooth and shiny [2]. Figure 1 presents characteristic signs of VLS. We can also observe erosions, blistering lesions, scars, adhesions and bruises. It may often be incorrectly recognized as a sign of sexual harassment, but one has to remember that those two cases do not exclude each other [23,30]. In the case of bruises in the intimate areas, the factors that should raise suspicion of sexual abuse are as follows: the appearance of lesions in older girls before puberty, poor response to the treatment, and the coexistence of other infection features that may suggest the presence of sexually transmitted diseases. Occasionally, the complication of dilated veins in the perineal area may also present a similar clinical picture [31,32,33,34,35]. The above-mentioned can result in labial resorption, adherence of opposing labia, covering of the clitoris, and narrowed vestibule of the vagina. Perianal involvement is a frequent finding in young girls who may present with constipation because of painful fissuring in this area. Dysuria can also result from fissuring. One single study indicated that 10 out of 15 (66%) girls with VLS presented with perianal lesions, and the incidence of perianal lesions is much higher in female patients, including children [23]. What is important is that the extent of the anogenital lesion does not correlate with the intensity of the symptoms; thus, small lesions may result in significant complaints [24].

In the case of eruption outside the anogenital area, these lesions can appear anywhere on the body—usually in the back, chest and breast areas, as white flat lumps that may coalesce to a form of larger foci with a shiny porcelain surface; sometimes they are surrounded by a purple halo. Less common sites include the mouth, face, scalp, hands, feet and nails [16]. As we mentioned before, the typical lesions are porcelain-white plaques, which, similarly to the genital lesions, may have follicular dells and areas of ecchymosis. It might be difficult to distinguish the lesions from those of morphoea. The clinical types of extragenital VLS include an extensive bullous form [19,21] and annular, blaschkoid and keratotic variants [22]. Koebnerization is very common at extragenital sites, arising at pressure points, old surgical and radiotherapy scars, and sites of trauma, including urostomies [18].

## 5. Complications in the Course of Vulvar Lichen Sclerosus

### Vulvar Squamous Cell Carcinoma (vSCC)

Vulvar lichen sclerosus, apart from the symptoms of the active phase of the disease, may be associated with various complications in the later years of life. One of the most serious is an increased risk of vulvar squamous cell carcinoma (vSCC). However, this refers only to genital forms of VLS [36]. In the entire population of VLS patients, 2.6–6.7% will undergo neoplastic transformation, which is the result of chronic inflammation, altered expression of the p53 oncogene, and oxidative stress [37,38]. Further, vSCC is a rare gynecological neoplasm with two main forms—related and unrelated to the HPV infection. Although VLS is not treated as a precancerous condition, mutations within the genes of the TP53 sequence are observed in its course. This confirms the clonal relationship between VLS and vSCC [23,38]. The analysis of the results of the histopathological examination of vulvar squamous cell carcinoma shows that 60% of cases develop on the basis of vulvar lichen sclerosus [36]. In the literature, we can also find reports of an increased incidence of anal and genital cancer in familial VLS, which may justify family screening for vulvar lichen sclerosus, in order to effectively prevent genital cancers [23]. A correct treatment of VLS is probably associated with a lower risk of neoplastic transformation; on the other hand, it should be remembered that the topical immunosuppression in the course of treatment with calcineurin inhibitors may increase the oncological risk [39]. It should be emphasized that in pediatric patients, the incidence of vSCC is lower than in the rest of patients with VLS, and it amounts to approx. 2.2%. This occurs due to the pathogenesis of the neoplasm and additional risk factors—mainly age. However, the study by Halonen et al. showed that the early age of VLS diagnosis is an additional risk factor for this complication [39]. In the available literature, individual cases of the early onset disease are described. Cario et al. described the case of an 18-year-old female patient, while Wallace described the case of a 25- and 35-year-old female patient with vSCC, diagnosed with vulvar lichen sclerosus in childhood [39].

In patients with vulvar lichen sclerosus, scars and deformities of the affected areas are observed more often. The available study results are not unequivocal, and they define the frequency of these ailments from 20 to 50%. Cooper et al. found that scarring is more likely in women with VLS than in girls, and the probability is 79% and 20%, respectively [39]. It has not been determined whether the early treatment initiation improves the prognosis for the occurrence of these complications. The described changes may coexist with chronic pain and itching, significantly worsening the quality of life. In sexually active patients, penetration difficulties and dyspareunia may appear [23,39].

The narrowing of the vulval vestibule may occur as a result of the fusion of the labia. If this produces the inability to have sexual intercourse or causes problems with urination, surgery may be necessary. Topical steroids and dilators can help in the postoperative period, by preventing re-deformation. Occasionally, the clitoral adhesions result in the formation of a painful pseudocyst, which is also a condition requiring surgical intervention [36].

As a result of the inflammatory condition in the course of VLS, sensorial disturbances in the vulva may develop and lead to vulvodynia. These symptoms may persist during and after the treatment. Pain sensation is not reduced by treatment with glucocorticosteroids. In such a case, the patient should be offered a treatment aimed at neuropathic pain [36].

## 6. Diagnosis

The differentiation of symptoms in the course of vulvar lichen sclerosus causes great diagnostic difficulties. In only 16% of cases, girls are diagnosed with vulvar lichen sclerosus in the initial stage of the disease. Changes in the course of VLS may imitate the clinical symptoms of many dermatoses. The differential diagnosis of vulvar lichen sclerosus includes lichen simplex chronicus, lichen planus, eczema, psoriasis, atopic dermatitis, seborrheic skin lesions, vitiligo, sexual harassment, vulvar injuries, and linear IgA disease of the childhood (chronic bullous dermatosis) in cases when VLS presents with bullae [40].

### 6.1. Differential Diagnosis

Lichen simplex chronicus (LSC)—can develop in the background of long-lasting primary lichen sclerosus. It is much more common in adults than in girls. The lesions are in the form of sharply delimited, exfoliating plaques, which are characterized by the so-called “saw teeth”;Lichen planus—lesions often affect the oral and vaginal mucosa, and are typical of peri and postmenopausal women;Atopic dermatitis—a history of exposure to external allergens should always be kept in mind. The characteristics of atopic dermatitis are scaly, irregularly shaped lesions that appear on the erythematous surface;Psoriasis—the typical location of lesions are the elbows and knees, and red–brown lesions are covered with strongly adherent silvery scales;Vitiligo—is most often confused with lichen sclerosus, due to the hypopigmentation; however, patients with vitiligo are typically asymptomatic and do not have evidence of vulvar structure atrophy;Skin mycosis—there are satellite changes around;Seborrheic skin lesions—occur mainly on the scalp and nasolabial folds and take the form of scaly, oily, erythematous, symmetrical spots [40].

In most cases, the diagnosis is made on the basis of a thorough medical history interview with the patient/guardian, and as the result of a physical examination. Vulvar lichen sclerosus is a clinical diagnosis, and a confirmatory biopsy is not always necessary when the typical clinical features are present. This is particularly true in children and men. However, the histological examination is recommended if there are atypical features or diagnostic uncertainty, and it is essential if there is any suspicion of a neoplastic lesion. It is important to remember that vulva biopsies are reserved for doubtful cases, especially when there is no improvement after treatment [23,36,40].

According to the guidelines of the British Dermatological Society, skin biopsy should be performed in the following cases (taking material for histopathological examination, with particular emphasis on hyperkeratotic lesions):The disease fails to respond to an adequate treatment, or an alternative/additional therapy with a potent topical steroid is to be implemented;There is an extragenital lichen sclerosus that has features mimicking the morphoea;There are pigmented areas to exclude abnormal melanocytic proliferation;There is a suspicion of neoplastic lesion. These are usually lesions with a persistent area of hyperkeratosis, erosion or erythema, or new warty or papular lesions. Several mapping biopsies may be required if there is an extensive abnormality. If there are any lesions that are highly suspicious of an SCC, the patient should be referred urgently to a gynecologist for the excision of the whole lesion for an adequate staging [36].

### 6.2. Histological Features

As we mentioned before, in children, a vulval biopsy is usually not performed, because it may be very traumatic for the child. It should be reserved only for cases with an uncertain diagnosis, and for those who fail to respond to treatments [41]. The typical histological features of VLS are orthohyperkeratosis, epidermal atrophy, basal cell degeneration, dermal hyalinization, and a band-like lymphocytic infiltrate [23].

## 7. Pharmacotherapy

In the natural course of vulvar lichen sclerosus, periods of remission and relapse are observed. In the absence of VLS remission symptoms in children until puberty, the prognosis is uncertain and may be associated with the lack of full recovery. The main goal of pharmacotherapy for vulvar lichen sclerosus is to alleviate the bothersome clinical symptoms, and prevent complications such as scars and adhesions [40]. Table 1 summarizes VLS treatment options with their effectiveness and side effects.

### 7.1. Topical Glucocorticosteroids

Topical glucocorticosteroids treatment with a 0.05% ointment containing clobetasol propionate constitutes the golden standard in the therapy of girls suffering from vulvar lichen sclerosus. According to the recommendations of the British Association of Dermatologists, it is recommended to use 0.05% clobetasol propionate ointment in combination with an emollient for three months, with the following gradual dose-reduction scheme during the therapy: once a day in the first month of treatment, then every second day for a month, and two times a week during the third month of treatment. Children diagnosed with vulvar lichen sclerosus should receive long-term care—with lesions controlled three and then six months after the start of the therapy [36,42]. Additionally, according to the studies results, a follow-up visit is recommended after four weeks of therapy, to assess the effectiveness and safety of the treatment and the occurrence of possible complications, such as stretch marks, atrophic skin changes, and secondary reproductive tract infections. The maximum allowable dose of clobetasol propionate is about 10 g per month [36,40,42,43]. According to the European guidelines for the treatment of vulvar lichen sclerosus, based on the EBM principles (evidence-based medicine principles), the proposed first-line treatment in girls is an ointment with 0.05% clobetasol propionate [23].

Cassey et al. compared the effectiveness of various VLS therapies—moderately strong corticosteroids (ointment with 1% hydrocortisone and 0.05% clobetasol butyrate) with corticosteroids with a very strong effect (ointment with 0.05% clobetasol propionate). The study group included 62 girls who were treated for three months with an ointment containing 0.05% clobetasol propionate. The stage of disease advancement was assessed at the time of diagnosis, and then the effectiveness of the applied treatment after 3, 6 and 12 months, from the beginning of the therapy and annually for four to eight years until puberty. A clear improvement in the clinical symptoms was observed in 96.8% of the patients, and in 72.6% of the girls a complete remission of the lesions was achieved after a three-month treatment period. Moreover, in the prospective evaluation (four-year follow-up) of the treatment effectiveness, 29.2% of the girls had a complete remission of VLS, and 37.5% of the study group required the use of the medicine less frequently than once a week. Comparing the obtained results in the study group of 62 girls with the control group of 31 girls (retrospective evaluation of the effectiveness of the therapy with moderately effective corticosteroids), only 32.2% achieved a complete remission of VLS symptoms. What is more, as many as 67.7% of the girls used the topical ointments chronically, at least twice a week, in order to relieve the bothersome symptoms of VLS. Based on the obtained research results, the authors recommend the ointment therapy with 0.05% clobetasol propionate as a first-line treatment in the pre-pubertal girls, due to its high effectiveness and chance for a complete recovery [44].

Continuous studies to establish the pharmacotherapy of VLS with the use of topical glucocorticosteroids suggest that in 65–100% of girls treated with the clobetasol propionate ointment, the vulvar lichen sclerosus symptoms resolved, while 20–70% had a complete remission, without the need for maintenance treatment in 55% of the girls [23].

Mashayekhi et al. conducted a detailed literature review of the possible treatments for vulvar lichen sclerosus in children, taking into account publications from 1971 to 2016. In 141 girls treated in a three-month cycle with an ointment with 0.05% clobetasol propionate, remission or significant improvement of the reported symptoms was achieved; however, in a long-term follow-up, some patients experienced a relapse. The maintenance treatment of VLS requires an individual approach to each patient, and is still a subject of much debate among clinicians. The authors emphasize the need for further research in this age group of patients, in order to establish long-term, effective maintenance treatment strategies to prevent disease recurrence and the remote effects of VLS—scars, adhesions or dyspareunia [45].

It is recommended to follow-up at least until puberty, and in the case of treatment-resistant or atypical vulvar lichen sclerosus, the adolescent patient should be referred to a specialized vulval disease clinic [36].

Treatment with clobetasol propionate ointment was considered to be safe in the group of girls. In the study by Cassey et al., erythema (12.9% of girls) and telangiectasia (19.4%) were observed in the study group of 62 girls [44]. The literature describes one case of Cushing’s syndrome in a six-year-old girl who was treated for eight weeks with strong topical corticosteroids—0.05% clobetasol ointment for vulvar lichen sclerosus. The risk factors for this extremely rare complication include the age of the adolescent patient (higher risk in younger children), the type of topical corticosteroid used, the duration of the treatment, the severity of the disease, and the condition of the skin. Therefore, it is extremely important to inform the child’s parent/guardian in detail about the type of treatment proposed, and the need for systematic monitoring of its effectiveness and possible side effects [46].

Ointment preparations are recommended in the treatment of vulvar lichen sclerosus in girls, due to possible side effects of using the creams, such as discomfort in the vulva/perineum and irritation [43].

### 7.2. Calcineurin Inhibitors—Tacrolimus, Pimecrolimus

According to the characteristics of the preparation, tacrolimus is a strong immunosuppressive medicine that, by bonding to a specific cytoplasmic immunophilin, inhibits the calcium-dependent signal transduction cascades in T lymphocytes, thus blocking the synthesis of, among others, interleukin-2, -3, -4, -5 and other cytokines. In addition, it inhibits the release of inflammatory mediators from skin mast cells, basophils and eosinophilia. It has been observed that in patients with atopic dermatitis during their treatment with tacrolimus, the skin lesions healing is associated with a decrease in Fc receptor expression in Langerhans cells and inhibition of their activating effect on T lymphocytes. However, the mechanism has not been fully elucidated [23].

The European guidelines for the treatment of vulvar lichen sclerosus, based on the principles of the EBM, emphasize that, according to research results, 0.03% tacrolimus ointment is considered to be an effective and safe form of VLS therapy. Additionally, the authors recommend the use of tacrolimus as a maintenance treatment (twice a week), which results in better control of the clinical course of the disease and a reduction in the frequency of relapses [23].

Li et al. conducted a study in 14 girls aged 4–11 years, who were treated with the 0.03% tacrolimus ointment. At the annual follow-up, the authors observed clinical improvement in all of the girls and a relapse in 80% of the girls receiving the treatment for 16 weeks, and only in 22% of the girls who received maintenance treatment with tacrolimus ointment twice a week for six months. On the basis of the obtained results, they concluded that the topical application of low-concentrated tacrolimus seems to be an effective and safe therapy in the treatment of vulvar lichen sclerosus in children, and that maintenance therapy (twice a week for six months) may reduce the risk of disease recurrence [47].

Ellis et al. assessed the effectiveness of maintenance therapy with tacrolimus in 46 girls with VLS. At the time of the follow-up (32 months on average), 93.3% of the girls regularly using tacrolimus ointment had no progression of vulvar lichen sclerosus. The authors recommend an individual approach to each adolescent patient, and the maintenance treatment with tacrolimus ointment in pre-pubertal girls in order to maintain the disease remission and prevent the remote complications of VLS [30].

Feito-Rodriguez described the case of a 10-year-old girl who was diagnosed with bacterial vaginosis, caused by the beta-hemolytic streptococci of group A, while treating the vulvar lichen sclerosus with tacrolimus ointment. The authors hypothesized that the immunosuppressive effect of tacrolimus may have an impact on the overlapping of bacterial vaginosis on the clinical picture of VLS, which is a rare phenomenon in the course of this disease [48].

In the European guidelines for the treatment of vulvar lichen sclerosus, we can also find data on the effectiveness of pimecrolimus in the treatment of VLS in pre-pubertal girls. The authors emphasize that it may offer an alternative to topical corticosteroids [23].

The main side effect of treatment with calcineurin inhibitors is the burning sensation, which gradually subsides over time. This therapy has a better effect in treating the symptoms of skin atrophy. Both of the medicines are classified by the Food and Drug Administration as “black box”-labeled preparations, because their long-term topical use is associated with skin cancer and lymphoma. Both of the medicines are not recommended for use in children under two years of age [40,49].

### 7.3. Topical Sex Hormons

Topical sex hormones have not been tested for their efficacy and safety in the treatment of VLS in children. In adult women, creams with 0.1% estrogen show a supportive effect, especially in the cases of estrogen deficiency, when the symptoms of vulvar lichen sclerosus are accompanied by vaginal dryness, dyspareunia and vaginal prolapse. Patients with early onset VLS may benefit from using 8% progesterone ointment. However, the conducted studies emphasize the advantage of ointment therapy with clobetasol propionate in this group of patients [23]. There is a single report of the use of progesterone ointment in a pediatric patient, after which the pruritus was completely reduced, while the persistent skin lesions were still present [23,45]. Ointments with 2% testosterone propionate are not recommended, due to low effectiveness and unacceptable side effects, such as clitoral enlargement, hirsutism, and acne [23].

### 7.4. Retinoids

The topical effect of retinoids on VLS symptoms may be explained by their influence on CD44 expression increase in the epidermis. As Kaya described in his research, in VLS lesions, epidermal CD44 expression is reduced or absent, both at the protein and mRNA levels; this is related to the build-up of hyaluronan. In the study of Virgili et al., patients were treated with 0.025% tretinoin once a day, 5 days a week for a year. The topical use of retinoids resulted in the resolution of symptoms and the disappearance of skin lesions. Further, 76% of the patients no longer suffered from itching. The reduction in burning and pain sensations during sexual contacts was similar. Retinoids, however, are not recommended for monotherapy treatment, but rather as an adjunctive treatment during topical glucocorticosteroids therapy. Oral therapy is not recommended, due to its numerous side effects (dry skin, soreness and peeling of the lips with cracking, soreness and bleeding of the inner nose, dry eyes, fragile skin, and minor skin infections) [23].

### 7.5. Cyclosporine

The literature describes the case of a seven-year-old female patient who was successfully treated with cyclosporine, dosed twice a day for eight months. After two weeks, the pruritus disappeared and the remaining symptoms—including the skin atrophy—resolved after the cycle was over. There were also no side effects observed [45].

### 7.6. Phototherapy

All of the studies with the use of phototherapy in the treatment of vulvar lichen sclerosus were performed in adult patients. There are studies documenting the effectiveness of this form of treatment in patients with VLS, also with genital lesions. PUVA (psoralen ultraviolet light A) mainly reduces the pain and burning sensation, while it is less effective in reducing itching. This treatment is well tolerated and no significant side effects have been reported. However, in randomized trials comparing the use of phototherapy with the topical administration of clobetasol propionate, glucocorticoids were found to be superior. Much less research concerns the UVB rays (ultraviolet B rays), but available data indicate a positive therapeutic effect. Phototherapy is not recommended as the first-line treatment of VLS, but patients with no expected response to corticosteroids may benefit from its use [23].

### 7.7. Vitamins D, A and E

Among the treatment adjuvants, the advantage of creams with vitamin E over emollients was not noticed. They provide a similar feeling of relief after their application, but it cannot be clearly stated whether it is a direct effect of these agents or the treatment with local glucocorticosteroids [23].

Better results are observed with oral vitamin supplementation, which is especially important for the proper functioning of the skin. The active form of vitamin D influences cellular differentiation, proliferation, and regulates immunity. Oral administration of calcitriol, at a daily dose of 0.5 micrograms, showed efficacy in the treatment of VLS in one patient. The lesions diminished and the skin density improved, but a side effect in the form of hypercalciuria was observed [23].

Vitamin A also has an impact on cell proliferation, while vitamin E has an antioxidant effect. The conducted research assumed the use of both vitamins in therapeutic doses with the simultaneous withdrawal of topical treatment. In 91% of patients, the disease symptoms remission was achieved, and it lasted for the next 12 months in 90% of cases. However, additional data are needed to assess the usefulness of vitamin supplementation in the treatment of VLS [23].

### 7.8. Treatment Failure

If treatment with topical corticosteroids appears to fail to bring VLS under control, then it is important to consider the following:Is noncompliance an issue? Sometimes patients may be alarmed at the contents of the package leaflet warning against the use of topical corticosteroids in the anogenital area. It is important to ensure that the treatment is being applied in an adequate amount and in the correct location;Has the correct diagnosis been made? If a biopsy was not done previously, it should be considered, to exclude differential diagnoses, including LP, mucous membrane pemphigoid or genital intraepithelial neoplasia;Are there any additional superimposed problems, such as the development of a contact allergy to the medication, or herpes simplex infection? Some patients can have VLS and psoriasis simultaneously, which may be more difficult to control [25,26];Patients with hyperkeratotic VLS often require further treatment and should be referred to a specialized clinic. The use of systemic retinoids may be considered in this group;Has the patient developed vulvodynia/penodynia? If the VLS has been successfully treated, but the patient remains symptomatic, often with burning or pain sensation being a predominant symptom rather than itch, vulvodynia/penodynia always has to be considered [36];

## 8. Conclusions

Lichen sclerosus is a chronic condition commonly affecting pre-pubertal females; it can be asymptomatic or associated with vulvar complaints. It can result in a decrease in the life quality, along with vulvar atrophy and scaring. Treatment is usually based on high-potency corticosteroids. A long-term follow up is recommended.

## Figures and Tables

**Figure 1 ijerph-18-07153-f001:**
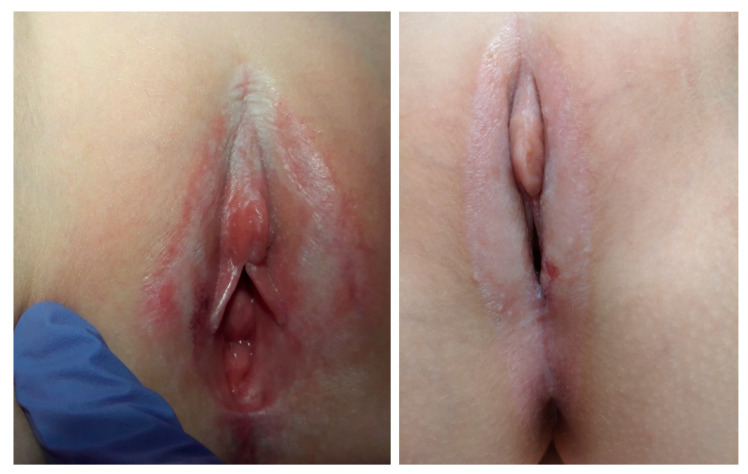
Classic vulvar lichen sclerosus in young girls (two cases)—own material.

**Table 1 ijerph-18-07153-t001:** Management for pediatric vulvar lichen sclerosus.

Treatment	Effects	Side Effects
High-potency corticosteroids	0.05% ointment containing clobetasol propionate—“golden standard”; 65–100% improved; complete reversal of signs in 20–70% (55% are without continuous treatment); treatment well tolerated.	Prolonged use of topical steroids can be associated with: thinning of the dermis; secondary superimposed infections; erythema; rarely hypothalamic–pituitary–adrenal axis suppression.
Calcineurin inhibitors—tacrolimus, pimecrolimus	Tacrolimus 0.03% ointment: complete response in 79% after 10 months; individual approach to each adolescent patient; maintenance treatment necessary. Pimecrolimus: effective in majority (relief of itch); no effect on sclerosis.	side effects of TCIs included stinging and burning; concern for the use of TCIs stems from the intrinsic malignant potential that TCIs may increase the risk of SCC development in patients with LS especially with long-term use (not recommended for use in children under two years of age).
Retinoids	not recommended for monotherapy treatment; the resolution of symptoms and disappearance of skin lesions; 76% of patients no longer suffered from itching.	No report.
Topical sex hormons	No report in children.	No report.
Cyclosporine	In patients with refractory VLS with symptomatic improvement and decrease in erythema and erosions after one month of therapy.	no side effects observed; limited data.
Phototherapy	No report in children.	No report.
Vitamins D, A and E	Additional data are needed to assess the usefulness of vitamin supplementation in the treatment.	No report.

## Data Availability

Not applicable.

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
