# Peer review of "Pediatric Vulvar Lichen Sclerosus—A Review of the Literature"

_ijerph, 2021, doi:10.3390/ijerph18137153_

Round 1

Reviewer 1 Report

It is a well written manuscript that reads very well. The authors tried to present an unusual health issue in an innovative way and have collected information using a wide range of academic literature. There is evidence of critical engagement with the relevant scholarly literature.

Author Response

Dear Reviewer,

We would like to express our gratitude for the review of the article and for the positive opinion.

Reviewer 2 Report

This is an interesting and important study. I suggest minor revision as follows.

  1. VLS and LS were mixed in the study. Consistent writing is required.
  2. presented review is your experience? or review of previous study? clarify this point and add references of prevous study.

Abstract

  1. Figure 8 was not described in manuscript (pg1 line 12)
  2. Change "vulvar LS" into "VLS" (pg1 line 12)

Introduction

  1. Add reference in end of sentence (pg1 line 22-23)

        - "···spreads to the skin of the anus [reference]."

  1. LS was not described in manuscript (pg1 line 26)
  2. GPs was not described in previous sentence (pg1 line 27)
  3. Add implication or contribution of this review (pg1 line 30)

Epidemiology

  1. Add reference at end of sentence (pg1 line 33)

       - "···and the pre-pubertal girls. [reference]."

  1. Study of Bleeker et al. is not proper for epidemiology part. Refer study "Vulvar Lichen Sclerosus et Atrophicus" and provide epidemiological information of VLS (pg1 line 40-41)

Etiopathogenesis

  1. Generally, references were omitted in Immunological Theory part (pg2 53-60), Hormonal Theory (pg2 line 74-82), and Trauma Theory (pg2 line 86-89)
  2. HRT was not described in manuscript (pg2 line 83)
  3. ACE was not described in manuscript (pg3 line 106)

Complications in the Course of Vulvar Lichen Sclerosus

  1. Change "2,6–6,7%" into "2.6-6.7%" (pg4 line 157)
  2. Change "VSCC" into "vSCC" (pg4 line 159) or change "vSCC" into "VSCC" (pg4 line 153)
  3. Change "2,2%" into "2.2%" (pg4 line 172)
  4. Change "vSSC" into "vSCC" (pg4 line 177)

Pharmacotherapy

  1. EBM was not described in manuscript (pg6 line 271)
  2. present details of side effects (pg8 line 390)
  3. PUVA was not described in manuscript (pg9 line 400)
  4. UVB was not described in manuscript (pg9 line 404)

Author Response

This is an interesting and important study. I suggest minor revision as follows.

  1. VLS and LS were mixed in the study. Consistent writing is required.
  2. presented review is your experience? or review of previous study? clarify this point and add references of prevous study.

Abstract

  1. Figure 8 was not described in manuscript (pg1 line 12)
  2. Change "vulvar LS" into "VLS" (pg1 line 12)

Introduction

  1. Add reference in end of sentence (pg1 line 22-23)

        - "···spreads to the skin of the anus [reference]."

  1. LS was not described in manuscript (pg1 line 26)
  2. GPs was not described in previous sentence (pg1 line 27)
  3. Add implication or contribution of this review (pg1 line 30)

Epidemiology

  1. Add reference at end of sentence (pg1 line 33)

       - "···and the pre-pubertal girls. [reference]."

  1. Study of Bleeker et al. is not proper for epidemiology part. Refer study "Vulvar Lichen Sclerosus et Atrophicus" and provide epidemiological information of VLS (pg1 line 40-41)

Etiopathogenesis

  1. Generally, references were omitted in Immunological Theory part (pg2 53-60), Hormonal Theory (pg2 line 74-82), and Trauma Theory (pg2 line 86-89)
  2. HRT was not described in manuscript (pg2 line 83)
  3. ACE was not described in manuscript (pg3 line 106)

Complications in the Course of Vulvar Lichen Sclerosus

  1. Change "2,6–6,7%" into "2.6-6.7%" (pg4 line 157)
  2. Change "VSCC" into "vSCC" (pg4 line 159) or change "vSCC" into "VSCC" (pg4 line 153)
  3. Change "2,2%" into "2.2%" (pg4 line 172)
  4. Change "vSSC" into "vSCC" (pg4 line 177)

Pharmacotherapy

  1. EBM was not described in manuscript (pg6 line 271)
  2. present details of side effects (pg8 line 390)
  3. PUVA was not described in manuscript (pg9 line 400)
  4. UVB was not described in manuscript (pg9 line 404)

Dear Reviewer,

First of all, we would like to express our gratitude for the review of the article and the constructive remarks regarding its content. We trust that, thanks to all the Reviewer’s suggestions, the manuscript has been substantially improved and it is suitable for publication. In the manuscript, we have highlighted changes using red colour. Responses to Reviewers comments are joined below in green colour.

Comments (C)

C1. VLS and LS were mixed in the study. Consistent writing is required.

Answer (A) 1 Thank you for comment. We apologize for inaccuracy. We have unified the abbreviations; VLS was used.

C2. presented review is your experience? or review of previous study? clarify this point and add references of prevous study.

A2. Vulvar Lichen Sclerosus is a field of our interest and the large study is ongoing. The final results will be published in the near future.

C3. Figure 8 was not described in manuscript (pg1 line 12)

A3. Thank you for this suggestion. The description was added to the manuscript.

C4. Change "vulvar LS" into "VLS" (pg1 line 12)

A4. Thank you for this comment. According to your suggestion, we have changed the nomenclature.

C5. Add reference in end of sentence (pg1 line 22-23)

        - "···spreads to the skin of the anus [reference]."

A5. Thank you for the suggestion. We have made appropriate change.

C6. LS was not described in manuscript (pg1 line 26)

A6. Thank you for this remark. According to your suggestion, we have changed the nomenclature. VLS is used instead of LS.

C7. GPs was not described in previous sentence (pg1 line 27)

A7. Thank you for this suggestion. We have explained the abbreviation.

C8. Add implication or contribution of this review (pg1 line 30)

A8. Thank you for the suggestion. We have added the missing part:

“In the literature there is good evidence for the high potency corticosteroids as initial treatment but the need for maintenance treatment and the most effective long-term management strategies are not established. Further well-conducted randomized controlled trials with long term follow-up in pediatric population are required to establish VLS treatment.”

C9. Add reference at end of sentence (pg1 line 33)

       - "···and the pre-pubertal girls. [reference]."

A9. Thank you for the suggestion. We have made appropriate change and added the reference.

C10. Study of Bleeker et al. is not proper for epidemiology part. Refer study "Vulvar Lichen Sclerosus et Atrophicus" and provide epidemiological information of VLS (pg1 line 40-41)

A10. Thank you for this comments. The study of Bleeker was removed. We have provided actual epidemiological information of VLS:

C12. Generally, references were omitted in Immunological Theory part (pg2 53-60), Hormonal Theory (pg2 line 74-82), and Trauma Theory (pg2 line 86-89).

A12. Thank you for this important consideration. We do apologize for the oversight. Following your suggestion, we have provided the missing references.

C13. HRT was not described in manuscript (pg2 line 83)

A13. Thank you for this suggestion. We have explained the abbreviation.

C14. ACE was not described in manuscript (pg3 line 106)

A14. Thank you for this remark. We have explained the abbreviation.

C15. Change "2,6–6,7%" into "2.6-6.7%" (pg4 line 157)

C16. Change "VSCC" into "vSCC" (pg4 line 159) or change "vSCC" into "VSCC" (pg4 line 153)

C17. Change "2,2%" into "2.2%" (pg4 line 172)

C18. Change "vSSC" into "vSCC" (pg4 line 177)

A15-18. Thank you for this comment. It is our neglect. We have revised the manuscript and corrected typing errors.

C19. EBM was not described in manuscript (pg6 line 271)

A19. Thank you for this remark. We have explained the abbreviation.

C20. present details of side effects (pg8 line 390)

A20. We have provided the lacking pieces of information in the manuscript:

“dry skin, soreness and peeling of the lips with cracking, soreness and bleeding of the inner nose, dry eyes, fragile skin and minor skin infections”

C21. PUVA was not described in manuscript (pg9 line 400)

A21. We have explained the abbreviation.

C22. UVB was not described in manuscript (pg9 line 404)

A22. We have described the abbreviation.

Hopefully, the corrected version would be suitable for publication.

Reviewer 3 Report

Drosdzol-Cop et al. gave a brief revew on pediatric vulvar Lichen Sclerosus. This manuscript is well-written and quite readable. Here are my suggestions:

  1. Abstract. As a review, the abstract should give a summary of the pediatric vulvar Lichen Sclerosus field. It not only includes the current knowledge but also points out unsolved questions. The abstract looks like it is not finished. The authors gave a very detailed introduction to pediatric vulvar Lichen Sclerosus. But I’d like to see some opinions from the authors, such as what the most important question in this area, or, what/why this review focus on.

  1. The Etiopathogenesis part is the weakest part of the whole review and Complications in the Course of Vulvar Lichen Sclerosus & Pharmacotherapy are the best parts. Thus, I suggest authors to say something about these two parts in the abstract and introduction.

  1. It will be great if the authors can add a figure to show the typical symptoms.

  1. Some figures for the Histological Features should be helpful.

  1. L157 ‘2,6–6,7%’ should be ‘2.6–6.7%’. There are many places in the whole manuscript.

  1. I’d like to see a table to summarize for all the medicines, their effects and side effects.

Author Response

Drosdzol-Cop et al. gave a brief revew on pediatric vulvar Lichen Sclerosus. This manuscript is well-written and quite readable. Here are my suggestions:

  1. Abstract. As a review, the abstract should give a summary of the pediatric vulvar Lichen Sclerosus field. It not only includes the current knowledge but also points out unsolved questions. The abstract looks like it is not finished. The authors gave a very detailed introduction to pediatric vulvar Lichen Sclerosus. But I’d like to see some opinions from the authors, such as what the most important question in this area, or, what/why this review focus on.

  1. The Etiopathogenesis part is the weakest part of the whole review and Complications in the Course of Vulvar Lichen Sclerosus & Pharmacotherapy are the best parts. Thus, I suggest authors to say something about these two parts in the abstract and introduction.

  1. It will be great if the authors can add a figure to show the typical symptoms.

  1. Some figures for the Histological Features should be helpful.

  1. L157 ‘2,6–6,7%’ should be ‘2.6–6.7%’. There are many places in the whole manuscript.

  1. I’d like to see a table to summarize for all the medicines, their effects and side effects.

Dear Reviewer,

First of all, we would like to express our gratitude for the review of the article and the constructive remarks regarding its content. We trust that, thanks to all the Reviewer’s suggestions, the manuscript has been substantially improved and it is suitable for publication. In the manuscript, we have highlighted changes using red colour. Responses to Reviewers comments are joined below in green colour.

Comments (C)

C1. Abstract. As a review, the abstract should give a summary of the pediatric vulvar Lichen Sclerosus field. It not only includes the current knowledge but also points out unsolved questions. The abstract looks like it is not finished. The authors gave a very detailed introduction to pediatric vulvar Lichen Sclerosus. But I’d like to see some opinions from the authors, such as what the most important question in this area, or, what/why this review focus on.

Answer (A) 1 Thank you for comment. We have revised the abstract and added the following pieces of information::

“This article reviews vulvar lichen sclerosus in children and provides evidence-based medicine principles for treatment in the pediatric population. Systematic search of the literature shows recurrence of VLS in children. Maintenance regimens deserve further consideration”

C2. The Etiopathogenesis part is the weakest part of the whole review and Complications in the Course of Vulvar Lichen Sclerosus & Pharmacotherapy are the best parts. Thus, I suggest authors to say something about these two parts in the abstract and introduction.

A2. Thank you for consideration. Following your suggestion, the missing data were added to the article in the abstract (as in the Answer 1) and in the introduction (“In the literature there is good evidence for the high potency corticosteroids as initial treatment but the need for maintenance treatment and the most effective long-term management strategies are not established. Further well-conducted randomized controlled trials with long term follow-up in pediatric population are required to establish VLS treatment.”)

C3. It will be great if the authors can add a figure to show the typical symptoms.

A3. Thank you for this suggestion. The figure was added to the manuscript.

C4. Some figures for the Histological Features should be helpful

A4. Thank you for this comment. Histological Features would be of interesting data. However, VLS is a clinical diagnosis and a confirmatory biopsy is rarely necessary. The article is designed for gynecologists not for pathologists. Therefore, we have decided not to include the figure of biopsy.

C5. L157 ‘2,6–6,7%’ should be ‘2.6–6.7%’. There are many places in the whole manuscript.

A5. Thank you for the suggestion. We are sorry for those neglects. The have revised the manuscript and made appropriate changes.

C6. I’d like to see a table to summarize for all the medicines, their effects and side effects.

A6. Thank you for this remark. According to your suggestion, we have added the table dealing with the medicines:

Table I. Management for paediatric vulvar lichen sclerosus

Treatment

Effects

Side effects

High Potency Corticosteroids

  • 0.05% ointment containing clobetasol propionate – “golden standard”
  • 65% -100% improved
  • complete reversal of signs in 20% - 70% (55% are without continuous treatment)
  • treatment well tolerated

Prolonged use of topical steroids can be associated with:

  • thinning of the dermis
  • secondary superimposed infections
  • erythema
  • rarely hypothalamic-pituitary-adrenal axis suppression

Calcineurin inhibitors - tacrolimus, pimecrolimus

Tacrolimus 0.03% ointment:

  • complete response in 79% after 10 months
  • individual approach to each adolescent patient
  • maintenance treatment necessary

Pimecrolimus:

  • effective in majority (relief of itch)
  • no effect on sclerosis
  • side effects of TCIs included stinging and burning.
  • concern for the use of TCIs stems from the intrinsic malignant potential that TCIs may increase the risk of SCC development in patients with LS especially with long-term use (not recommended for use in children under 2 years of age)

Retinoids

  • not recommended for monotherapy treatment
  • the resolution of symptoms and disappearance of skin lesions.
  • 76% of patients no longer suffered from itching.

No report

Topical sex hormons

No report in children

No report

Cyclosporine

In patients with refractory VLS with symptomatic improvement and decrease in erythema and erosions after one month of therapy

  • no side effects observed
  • limited data

Phototherapy

No report in children

No report

Vitamins D, A and E

Additional data are needed to assess the usefulness of vitamin supplementation in the treatment

No report

Hopefully, the corrected version would be suitable for publication.